# Earthworms and plants can decrease soil greenhouse gas emissions by modulating soil moisture fluctuations and soil macroporosity in a mesocosm experiment

Pierre Ganault[1,2,3,4]*, Johanne Nahmani[2], Yvan Capowiez[5], Nathalie Fromin[2¤], Ammar Shihan[2], Isabelle Bertrand[6], Bruno Buatois[2], Alexandru Milcu[2,7]

1 ECODIV, INRAE, Normandie Université, UNIROUEN, Rouen, France, 2 CEFE, Univ Montpellier, CNRS, EPHE, IRD, Montpellier, France, 3 German Centre for Integrative Biodiversity Research (iDiv) Halle-Jena-Leipzig, Leipzig, Germany, 4 Institute of Biology, Leipzig University, Leipzig, Germay, 5 INRAE, UMR 1114 EMMAH, INRAE/Université d'Avignon, Site Agroparc, Avignon, France, 6 UMR Eco&Sols, CIRAD, INRAE, IRD, Montpellier SupAgro, Université de Montpellier, Montpellier, France, 7 Montpellier European Ecotron, Univ Montpellier, CNRS, Campus Baillarguet, Montferrier-sur-Lez, France

¤ Current address: CNRS– 7 rue du Four Solaire, Odeillo, France

* pierre.ganault@univ-rouen.fr

**Data Availability Statement:** The datasets and R code of the current study is currently available on

## Abstract

Earthworms can stimulate microbial activity and hence greenhouse gas (GHG) emissions from soils. However, the extent of this effect in the presence of plants and soil moisture fluctuations, which are influenced by earthworm burrowing activity, remains uncertain. Here, we report the effects of earthworms (without, anecic, endogeic, both) and plants (with, without) on GHG ($CO_2$, $N_2O$) emissions in a 3-month greenhouse mesocosm experiment simulating a simplified agricultural context. The mesocosms allowed for water drainage at the bottom to account for the earthworm engineering effect on water flow during two drying-wetting cycles. $N_2O$ cumulative emissions were 34.6% and 44.8% lower when both earthworm species and only endogeic species were present, respectively, and 19.8% lower in the presence of plants. The presence of the endogeic species alone or in combination with the anecic species slightly reduced $CO_2$ emissions by 5.9% and 11.4%, respectively, and the presence of plants increased emissions by 6%. Earthworms, plants and soil water content interactively affected weekly $N_2O$ emissions, an effect controlled by increased soil dryness due to drainage via earthworm burrows and mesocosm evapotranspiration. Soil macroporosity (measured by X-ray tomography) was affected by earthworm species-specific burrowing activity. Both GHG emissions decreased with topsoil macropore volume, presumably due to reduced moisture and microbial activity. $N_2O$ emissions decreased with macropore volume in the deepest layer, likely due to the presence of fewer anaerobic microsites. Our results indicate that, under experimental conditions allowing for plant and earthworm engineering effects on soil moisture, earthworms do not increase GHG emissions, and endogeic earthworms may even reduce $N_2O$ emissions.

the first author GitHub repository at the following link https://github.com/PGanault/EarthwormPlantGHG.

**Funding:** The author(s) received no specific funding for this work

**Competing interests:** The authors have declared that no competing interests exist.

## Introduction

Soil invertebrates strongly influence organic matter dynamics, nutrient cycling, and water and gas fluxes through their feeding activity, movement through soil layers, and interactions with other organisms [1–3]. Among them, earthworms are essential and promote soil fertility and plant productivity [4, 5], resulting in a 25% increase in crop yield [6]. Conversely, the presence of earthworms may increase $CO_2$ emissions by 33% and $N_2O$ emissions by 42% [7], which is notable because $N_2O$ has 265 times the global warming potential of $CO_2$ [8]. However, the microbial processes causing greenhouse gas (GHG) emissions are extremely complex and involve numerous interactions between earthworms, microbial communities, plants, and soil water and aeration status, determining C and N mineralization or stabilization [9–14]. Simultaneously, exploring the complexity of these interactions and mechanisms mentioned above in experimental settings poses a challenge. Most existing studies have focused on scenarios without plants and at constant soil moisture levels that favor earthworm and microbial activity [7]. This limits our understanding of how soil moisture fluctuations, modulated by earthworms and plants and their interactions, affect $N_2O$ and $CO_2$ emissions [7, 15].

Soil water content (SWC) is a long recognized key factor that explains up to 95% of GHG emissions [16, 17], as it drives microbial processes such as respiration, denitrification, and nitrification that produce GHG [9, 18, 19]. Indeed, SWC determines gas and nutrient diffusion and hence the availability of oxygen, nitrate, ammonium and carbon to microorganisms, thereby modulating their activity. Under anoxic conditions, at high SWC, $N_2O$ emissions are the highest, mostly due to denitrification, while aerobic conditions favor $N_2O$ emissions by nitrification [20, 21]. Similarly, a substantial body of evidence has shown that carbon substrate limitation occurs in drier conditions, while oxygen limitation occurs under conditions close to water saturation, with optimal conditions for respiration and hence $CO_2$ emissions at intermediate levels of SWC [22, 23]. Soil moisture fluctuations (e.g., drying–rewetting cycles) can also affect the proportion of nitrogen denitrified into $N_2O$ or $N_2$, thus modulating the $N_2O/N_2$ ratio that will be emitted into the atmosphere [24, 25]. Therefore, keeping soil moisture constant, as in most existing experiments, limits the occurrence of these mechanisms.

Earthworm feeding and burrowing activity strongly influence carbon and nutrient dynamics and gas and water fluxes in the soil profile. Earthworms' fresh casts are richer in organic matter and water than bulk soil and can promote microbial respiration, while the anoxic conditions in earthworm digestive tracts promote the growth of denitrifying bacteria, two mechanisms leading to higher $CO_2$ and $N_2O$ emissions, respectively [7, 26, 27]. Moreover, earthworms' aging casts can constitute carbon stocks by the physical protection of particulate organic matter when mixed with the mineral fraction and the increase in microbial necromass, which constitutes another form of stable carbon stock [12, 26, 28–30]. Earthworms also have indirect impacts by burrowing into the soil, changing soil macroporosity and affecting air and water fluxes, soil moisture, soil compaction, and $CO_2$ and $N_2O$ diffusivity [31]. The effect of earthworms on GHG emissions therefore depends on the balance between these mechanisms and whether they may occur in the experimental set-up.

As primary producers, plants control organic matter quantity and quality in soils [32]. Root exudates contribute more to carbon stabilization than plant litter because the production of labile compounds (e.g., simple sugars) in the mineral soil favors the microbially driven formation of mineral-associated organic matter [14, 33, 34]. Inputs of root-derived C substrates can lead to high transient $O_2$ demand and can cause suboxic microsites in the rhizosphere, thus favoring denitrification [9, 35]. Conversely, plants compete with microbes for nitrogen acquisition, reduce SWC by transpiration, modify soil porosity by root growth [36, 37], and thus can change the preponderance of the controlling $N_2O$ emission processes (nitrification,

denitrification) [9, 38]. Simultaneously, the action of earthworms burrowing through the soil profile increases soil aeration, water drainage and possibly soil drying [39–42], thus potentially creating conditions that are less favorable for denitrification and $N_2O$ emission [15, 43]. The above mechanisms all interact, as earthworms promote plant growth by increasing nutrient availability to plants and by consuming roots [4–6]. To the best of our knowledge, no study has investigated earthworm-mediated soil moisture variation effects on GHG emissions. Furthermore, the vast majority of existing experimental studies have used micro/mesocosms with sealed bottoms, which impedes water drainage and the modulation of water infiltration via earthworm burrows and root growth that would typically occur in realistic field conditions. Two studies that evaluated the impacts of soil moisture fluctuations on GHG in the presence of earthworms, although without plants, showed that cumulative $N_2O$ and $CO_2$ emissions were reduced in the presence of earthworms [15, 43].

Allowing earthworm burrowing activity to influence SWC could aid in understanding the varying effects of different earthworm species and ecological categories on earthworm-mediated GHG emissions. Indeed, lumbricid earthworms are broadly classified into three main ecological categories based on their feeding and burrowing characteristics: (1) anecic species that feed on surface litter by pulling it into permanent vertical burrows and creating surface casts, (2) epigeic species that live and feed in surface litter, making very few nonpermanent burrows and (3) endogeic species that live in the soil, feed on roots and soil organic matter, and make numerous nonpermanent burrows [44, 45]. Hence, anecic earthworms are likely to have a stronger impact on organic matter redistribution and water fluxes [38]. Conversely, endogeic and epigeic earthworms may primarily influence organic matter redistribution in the soil and at the surface, respectively, with epigeic earthworms having a lesser impact on water fluxes [46]. There is evidence that $CO_2$ and $N_2O$ emissions depend on the earthworm ecological category, with significantly higher emissions for the anecic group [7], but the net balance between mineralization and stabilization over time was reported to be highly variable within each category [47]. Additionally, whether this finding holds in the presence of plants and soil moisture fluctuations remains to be tested.

In this study, we assessed the impact of four levels of earthworm treatments (one endogeic species, one anecic species, both anecic and endogeic species, and a control without earthworms) and two levels of plant treatments (with or without a model grass species). We used a full factorial design ($4 \times 2 = 8$ treatment combinations, with 7 replicates each) during a three-month greenhouse mesocosm experiment simulating a simplified agricultural setup. The experiment involved simulating two drying–wetting cycles, and to facilitate the earthworm engineering effect on soil water infiltration and status, the mesocosms were designed to enable effective water drainage and escape via percolation. We measured weekly $CO_2$ and $N_2O$ fluxes, aboveground plant biomass, litter cover and multiple soil parameters representing potentially relevant predictors of GHG emissions, including soil nitrogen and water status, microbial biomass and respiration, denitrification potential and multiple metrics of soil macroporosity, using X-ray tomography. We hypothesized that 1) $CO_2$ and $N_2O$ emissions will be lower in the presence of earthworms relative to controls, as increased carbon and nitrogen mineralization will be offset by the drier and more aerated conditions due to the earthworm soil engineering effect (burrowing) on water drainage, 2) plant presence will reduce $N_2O$ emissions due to nitrogen and water uptake but will increase $CO_2$ emissions due to increased carbon substrates entering the soil via rhizodeposition, and 3) differences in $N_2O$ and $CO_2$ fluxes among the two earthworm ecological categories will be mediated by the burrowing patterns affecting soil (macro) porosity and water status, with lower emissions with higher microporosity expected, as these conditions increase the volumes of aerobic and dryer sites.

## Materials and methods

### Soil and biological material

The soil, classified as a gleyic luvisol, was excavated from a field margin adjacent to a wheat–corn–alfalfa rotation at the EFELE experimental site (Northwest of France, 8˚05′35.9"N, 1˚48′53.1"W) belonging to the long-term observatories SOERE-PRO-network. Only soil from the upper 0–30 cm layer was used in this experiment. The soil was composed of 14.6% clay, 72.1% silt and 13.3% sand, with a pH of 6.14 and a volumetric water content at field capacity of 39.2% (Soil Analysis Laboratory, INRA Arras, France). The soil contained 1.5% total organic matter, 0.84% carbon, and 0.1% nitrogen, with a C:N ratio of 8.4. The mesocosm setup aimed at reproducing the agricultural field from which it was sampled by using its soil and adding locally present earthworms and a plant species commonly used as a model system for cereal grass.

Adult individuals of *Lumbricus terrestris* L. were supplied by Wurmwelten Company (Dassel, Germany) and weighed 4.8 ± 1.3 g fresh weight on average. Adult individuals of *Aporrectodea icterica* Savigny were harvested from a pesticide-free orchard in Avignon by manual digging and weighed 0.4 ± 0.2 g fresh weight on average. The earthworms were kept in their original soil for 3 days at a temperature of 14 ± 2˚C and then placed in a mixture of the original soil and the experimental soil for one week at 8˚C in the dark before the onset of the experiment.

*Brachypodium distachyon* L. was the plant species selected for this study due to its small size and short life cycle of less than 3 months [48] and because it is frequently used in controlled environment experiments [49]. The seeds of the wild-type variety (Bd 21 WT) were supplied by Observatoire du Végétal, INRAE Versailles (Paris, France). After one week of germination in seedling trays in vermiculite, four seedlings were planted in each mesocosm outside the central cylinder that was introduced as a base for flux measurements of greenhouse gases (see S1 Fig). During the first 3 weeks, any dead seedlings were replaced. The experiment ended with a final destructive harvest.

### Mesocosm design and experimental treatments

The mesocosms consisted of PVC tubes 16 cm in diameter and 37 cm in height (S1 Fig). Each mesocosm was filled up to 3 cm from the brim with 9.2 kg of soil already containing 10% gravimetric water content, sieved to 2 mm, and compacted to a bulk density of 1.21 g cm$^{-3}$. The mesocosms were sealed at their base with a 1 mm mesh followed by a PVC lid pierced with 5 holes (1 cm in dia.), which allowed the drainage of surplus water out of the mesocosm. A transparent plastic film 10 cm in height was attached around the top perimeter of the mesocosms to prevent the earthworms from leaving them. As a previous meta-analysis indicated that earthworm effects on plant growth are more prevalent in the presence of crop residues, which serve as a food resource for soil biota [6], the soil surfaces of all mesocosms were covered with a 4 g litter mixture (2.2% N, C/N = 24) consisting of 1.3 g dry weight *Medicago truncatula* Gaertn. shoots and 2.7 g dry weight of *Zea mays* L. leaves, the equivalent of organic residue inputs of 1060 kg C ha$^{-1}$ and 44 kg N ha$^{-1}$.

The mesocosm experiment presented in this study included four levels of earthworm treatments (henceforth Ew): a control without earthworms, an anecic earthworm species (*L. terrestris*) with two individuals weighing 9.6 ± 1 g fresh weight (FW) on average per replicate, an endogeic earthworm species (*A. icterica*) with 7 ± 1.1 individuals weighing 2.9 ± 0.1 g FW on average per replicate, and a mixture of both species with one *L. terrestris* individual (4.9 ± 0.9 g FW biomass) and 5 ± 1.6 *A. icterica* individuals (1.7 ± 0.5 g FW biomass) per replicate. The

earthworm FW biomasses were the equivalent of 480, 145 and 330 g m$^{-2}$ for the anecic, endogeic and both earthworm treatment levels, respectively, which were 2- to 3-fold higher than that in the field of origin of the soil, where earthworm total biomass varied from 98 to 135 g m$^{-2}$. For *L. terrestris*, adding the proportional equivalent of the field biomass would mean introducing just one individual. Given the associated risk that the escape or death of this single individual could jeopardize the treatment, we chose to add two adult individuals of *L. terrestris*. As a result, the overall biomass was roughly double what is typically found per m^2 in agroecosystems. Note that *L. terrestris* was more recently reclassified as a species displaying traits belonging to both anecic and epigeic species [36], but in the vast majority of literature, it is considered an anecic species. The earthworm treatments were factorially crossed with two levels of plant (*B. distachyon* L.) treatments, i.e., with and without plants, with 7 replicates per treatment combination for a total of 56 mesocosms.

The experiment was conducted in a greenhouse kept at temperatures ranging between 20 and 23˚C during the day and 18–20˚C during the night with an air relative humidity of 80%. The natural light was supplemented during daytime by artificial lighting for 12 hours per day using high-pressure sodium lamps. The mesocosms were divided into two blocks corresponding to their position on the north or south bench of the greenhouse, and their position within the block was randomly changed twice a week, limiting the bias of the position within the block. The experiment ran for 12 weeks between March and May 2017.

## Watering protocol

The watering protocol was specifically designed to include soil moisture fluctuations (analogous to what occurs in natural conditions) and to allow earthworm burrowing to affect SWC. At the beginning of the experiment, the mesocosms were watered with 1.7 L of reverse osmosis water using a laboratory dispenser (two sessions of 850 ml each), a volume sufficient to observe water draining out of the mesocosms from the pierced lids at the bottoms of mesocosms. Measurements of weight changes after 24 h were used to calculate the weights of the mesocosms at field capacity (knowing that the soil already contained 10% gravimetric water, i.e., $\sim 0.9$ L). Changes in total mesocosm weight hence allowed for the estimation of the SWC during the experiment and its expression in terms of % of field capacity. At field capacity, the mesocosm water-filled pore space (WFPS) can be estimated at 71.0 ± 2.5% on average (calculated as WFPS = water content/porosity; porosity = 1 - bulk density/particle density; particle density = 2.7 Mg m$^{-3}$). The mesocosms were exposed to a drying phase until one of them reached almost 50% of field capacity, which occurred after 6 weeks. The volume of water lost by the driest mesocosm was determined and then added (second watering in two sessions again) to all mesocosms to set them back to 100% of field capacity. A second drying phase was imposed until the penultimate week, when a third watering was performed with the same amount of water as supplied in the second watering. Following this method, all mesocosms experienced two drying–rewetting cycles (Fig 1A and 1B).

## Response variables

$CO_2$ and $N_2O$ emissions measurements were carried out weekly and during the weeks with watering, always 24 h after watering events, to capture eventual emission peaks. A static sampling chamber approach was used, following the recommendations of Rochette (2011) [50]. The sampling chamber (9 cm dia., 6 cm height, 370 ± 1 mL, S1 Fig) was equipped with a bung of silicone rubber for gas sampling at the top. During sampling, the chamber was placed on a circular collar (S1 Fig) that was inserted in the center of the mesocosms during the setup of the mesocosms, which allowed measuring soil $N_2O$ and $CO_2$ fluxes without disturbing the plants.

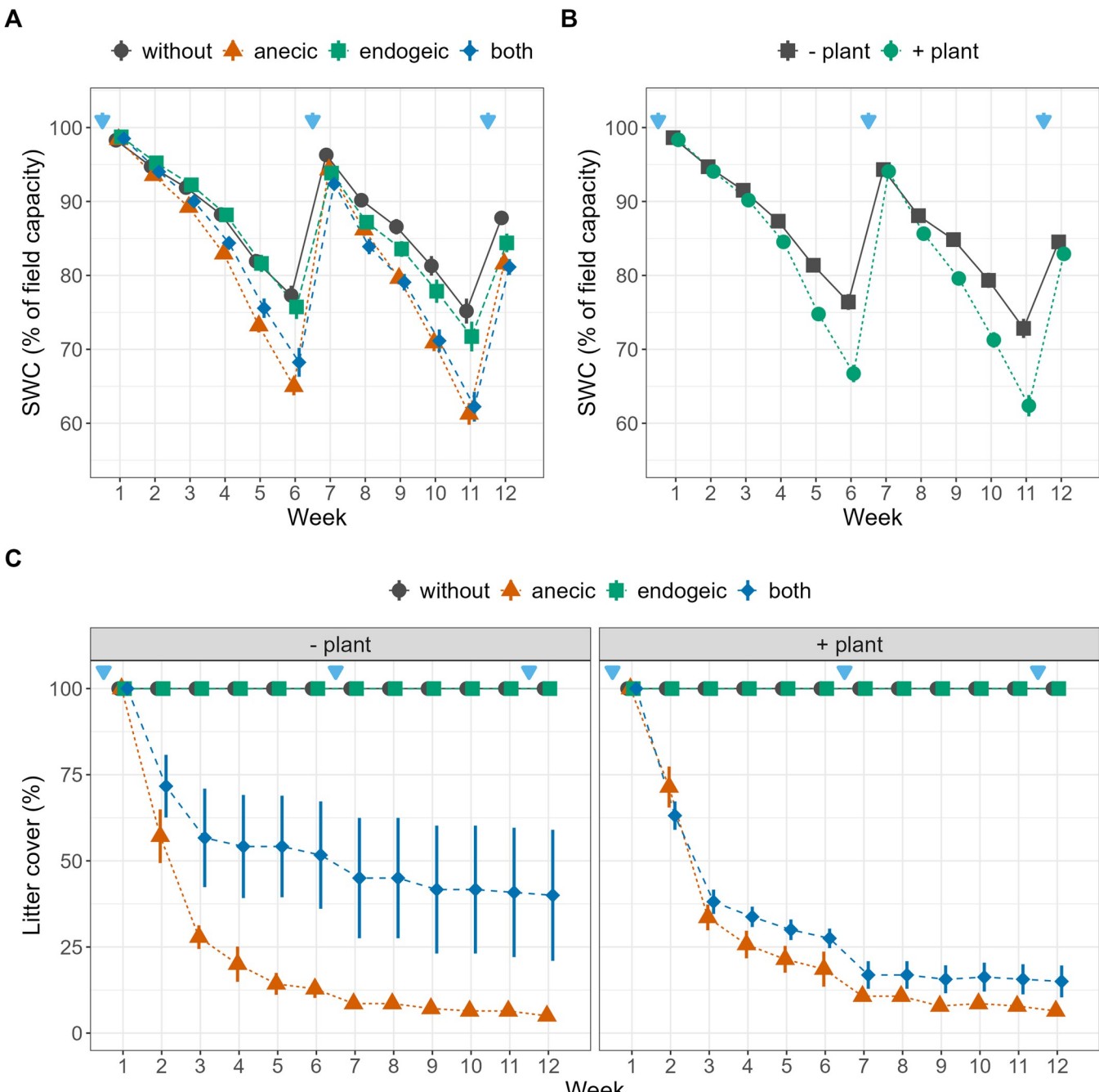

**Fig 1. Temporal dynamics of soil water content and litter cover.** (A,B) Temporal dynamics of soil water content (SWC) expressed as percentage of field capacity (C) and percentage of surface covered by litter as affected by the earthworm and plant treatments. Blue arrows represent watering events. Error bars represent ± 1 SEM. Different letters represent significantly different levels as estimated by Tukey's HSD post hoc test. The effects of earthworm treatments independent of plant treatments, and vice versa, are displayed when the Ew×Plant interaction is not significant (A,B).

The collar was inserted into the soil down to 3 cm and consisted of a frame that provided support but allowed access by earthworms and roots to the inner soil core thanks to two open windows. The aboveground part of the collar contained a gutter-like double-walled section/groove where the static chamber was placed during sampling. On the day of gas sampling, the

mesocosms were transported to the gas sampling laboratory by trolley, and the two blocks were measured over two days. Prior to sampling, 20 ml of distilled water was added to the groove to provide airtight sealing when placing the static chamber on the collar. $CO_2$ and $N_2O$ fluxes were measured at the Platform for Chemical Analysis in Ecology (LabEx CeMEB, Montpellier, France). $CO_2$ concentrations were measured with a gas chromatograph (MicroGC S-Series, SRA XXX Instruments, Marcy l'Etoile, France) using a catharometric detector, quantifying the gases on the basis of their thermal conductivity. $N_2O$ concentrations were measured by a gas chromatograph equipped with an electron capture detector (Varian CP-3800, Varian Inc., Palo Alto, USA). Air samples were taken at T0 (immediately after placing the chamber on the collar) and after 2 h to assess the changes in $CO_2$ and $N_2O$ concentrations. Previous tests were conducted after 1, 2, 3 and 4 hours, which revealed that gas accumulation was linear during this short time period. A volume of 0.2 mL was sequentially sampled for gas measurements via the silicone bung using a plastic syringe equipped with a 25G needle and was injected immediately into the gas chromatograph via a 1/32"PFA line. Concentration changes in the sampling chamber between T0 and T0+2 h were used to estimate the greenhouse gas emission rates and converted to g C-$CO_2$ and N-$N_2O$ m$^{-2}$ day$^{-1}$.

At the end of the experiment, the mesocosms were transported to the INRAE center of Nancy to analyze soil macroporosity by X-ray tomography using a medical scanner (Bright-Speed Exel 4, General Electric), with settings of 120 kV and 50 mA for the current and 0.625 mm width for each image. Images were transformed into 16-bit images and binarized (i.e., converted into black and white) using a fixed threshold value [51] because the different peaks (for the soil matrix and the porosity) were well separated [52]. Roots and associated pores could not be included in the analysis due to their smaller average size compared to the resolution of the scanner (0.4 mm per pixel). The burrow system was then characterized by computing the volume and number of burrows in four soil layers (L1 for 0–8.5 cm, L2 for 8.5–17 cm, L3 for 17–25.5 cm and L4 for 25.5–34 cm depth, S2 Fig) using ImageJ [41]. Drying–wetting cycles contributed to the formation of cracks, i.e., macropores resulting from physical processes (shrinkage, swelling [41]) notably in the topsoil layer (S2 Fig). We attempted to differentiate cracks from burrows according to the macropore circularity in 2D images because earthworm burrows are more circular than cracks. However, this method still identified burrows in the control mesocosms (without earthworms), with 25.4% of pores misidentified as burrows in the whole mesocosm and higher error in the first layer (32.2%) than in the bottom layer L4 (20.1%) (S2 Fig). Therefore, assuming that gas fluxes were influenced by the total porosity, regardless of its biological or physical origin, and due to the high correlations between the 15 different porosity variables (S3 Fig), although we investigated how the treatment affected the different types of soil pores (pores, burrows, and cracks), we decided to only use the total pore volume data (burrows and cracks) as a predictor in the models.

After the X-ray scan in Nancy, the mesocosms were transported back to Montpellier for the final destructive harvest. The proportions of earthworms found at the final harvest were 62% and 90% for *L. terrestris* and *A. icterica*, respectively. The proportions of recovered earthworms were likely affected by the mortality that occurred during the days of transport and storage for the X-ray scans (during which the temperatures and vibrations were not controlled), as several *L. terrestris* individuals were found freshly dead at harvest. However, the X-ray scans together with the litter mass loss dynamics (Fig 1C) provided strong evidence that the earthworms were active during the entire duration of the experiment. Litter cover was assessed weekly by a non-destructive visual estimation method with 5% intervals, as commonly performed for ground cover estimation [53, 54]. The observer bias of this method [55] was handled by having only one observer perform the estimations throughout the entire experiment. Other additional soil- and plant-related response variables were measured at the end of the experiment (S3 Fig). Soil

analyses were performed on a homogenized soil sample from the upper 10 cm of the meso-cosms inside the collar and sieved at 2 mm. Potential soil microbial denitrification enzymatic activity (DEA) was measured using the acetylene inhibition method, which measures total potential denitrification (as $N_2O$ and $N_2$) [56]. This is a complementary method to the fluxes measured during the experiment, which only measure the $N_2O$ emissions. The MicroResp$^{TM}$ method was used to determine the microbial metabolic quotient [57]. Approximately 0.39 g dry weight of soil was incubated in six replication wells with a solution of D-glucose (1.5 mg C $g^{-1}$ soil) and six replication wells with deionized water (for basal respiration) to reach 80% of the field capacity in 96-DeepWell Microplates (Fisher Scientific E39199). Cresol red gel detection plates were prepared as recommended by the manufacturer. After an initial two-hour pre-incubation at 25˚C in the dark, each DeepWell microplate was covered with a $CO_2$-trap microplate detection plate using a silicone gasket (MicroResp™, Aberdeen, UK). The assembly was secured with a clamp and incubated for four additional hours. The optical density at 590 nm (OD590) was measured for each detection well before and after incubation using a Victor 1420 Multilabel Counter (Perkin Elmer, Massachusetts, USA). Calibration relying on absorbance (OD590 readings) and $CO_2$ concentrations was performed using the gas chromatograph previously described. The final OD590 values were normalized using preincubation OD590 and converted as respiration rates expressed in µg C-$CO_2$ respired per $g^{-1}$ of soil per $h^{-1}$. The glucose-induced respiration rate was used to estimate the soil microbial C ($C_{mic}$, µg $C_{microbial}$ $g^{-1}$ dry soil) biomass [58]. Finally, the metabolic quotient (Met_Q) was determined as the ratio between basal respiration rates measured in the wells with water only and no C substrate (as a proxy of the microbial basal respiration) and $C_{mic}$. Soil mineral nitrogen was extracted from 10 g of freshly sampled soil with 40 mL of 1 M KCl solution. Nitrate and ammonium concentrations were measured by continuous flow spectrophotometry (SKALAR 3000 auto analyzer, Breda, The Netherlands). The plant shoot biomass was weighed after drying at 60˚C for three days. As the roots of *B. dystachyon* are extremely thin and fragile, it was not feasible to sample root biomass.

## Statistical analyses

Statistical analyses were performed using R version 4.0.2 (R Development Core Team, 2015) in RStudio version 1.3.959 (RStudio Team, 2015). Weekly time series of $CO_2$ and $N_2O$ emissions and SWC were analyzed with the "nmle" package version 3.1–145 [59] to perform repeated measures analyses using a generalized mixed-effects model to test the effects of earthworms, plants, SWC (for GHG emissions only) and sampling week and their interactions on gas fluxes. The identity (ID) of the mesocosm and its position in the blocks were used as random factors to account for temporal pseudoreplication and the effect of the position in the north or south bench in the greenhouse ("random = ∼ 1 | Block / ID"). To reach models that respected the assumption of homoscedasticity of the residuals, we tested the model fit with varIdent (for plant and earthworm experimental treatments), varPower (for SWC) and varExp (for SWC) weighting functions [60] and selected the most appropriate models based on maximum likelihood (ML) model comparison tests. A similar approach was used for the cumulative $CO_2$ and $N_2O$ data at the end of the experiment (estimated assuming constant emission rates between the weekly measurements), but without the sampling week among the fixed effects and the mesocosm ID in the random effects. For the later analysis, we used the mean of weekly SWC values, as this variable is arguably more relevant to the cumulative fluxes. The "r.squaredGLMM" function from the MuMIn package [61] was used to derive the proportion of the variation explained by the fixed factors (i.e., marginal $r^2$, $mr^2$) in mixed-effects models. The "multcom" package was used to perform Tukey's HSD (honestly significant difference)

multicomparison posthoc test, but note that this test does not include the random effects, and the results are occasionally not entirely in line with the fitted coefficients from the mixed-effects models.

Additional analyses were conducted aiming to link the multiple potential predictors (S3 Fig) measured at the end of the experiment and the $CO_2$ and $N_2O$ fluxes from week 12 (just before the experiment was stopped). As multiple response variables were measured to explore potential predictors, a method of best subset selection that penalizes model complexity (i.e., regularization) during estimation was needed. Regularization aims to significantly reduce the variance of the model as well as model overfitting by varying the lambda ($\lambda$) parameter, which tunes the level of penalization for the complexity of the model. This approach has been proven to be a viable option for estimating parameters in scenarios with small sample sizes and many collinear/correlated predictors. Here, we used a penalized regression method based on the minimax concave penalty (MCP) to select the best subsets [62] using the *ncvreg* package 3.11–1 [63]. This approach was combined with a 10-fold cross-validation procedure to derive the lambda parameter (also called the regularization rate), which minimizes the cross-validation error. We report the fitted coefficients and the coefficients of determination ($r^2$) at lambda values that minimize the cross-validation error. The subset variables with retained nonzero coefficients were then tested in the generalized mixed-effects models, which have the advantage of including random effects alongside the treatment factors.

## Results

### Soil water content

Over time, the SWC, expressed as a percentage of field capacity, exhibited variations due to the drying–rewetting cycles and the treatments involving plants and earthworms, which were reflected in the Plant×Week and Ew×Week interactions (Table 1 and Fig 1A and 1B). The SWC was significantly lower in the presence of plants during the last three weeks of both drying cycles, with 5% lower values in the absence of plants across the whole experiment (averaged over the earthworm treatments). The presence of anecic earthworms (with or without endogeic earthworms) led to significantly lower SWC values relative to the control during weeks three to six and eight to twelve. Averaged over the twelve weeks, the SWC values were 81.3% of field capacity in the presence of anecic earthworms, 81.7% in the presence of both earthworm species, 85.9% with endogeic earthworms and 87.0% in the control.

### Weekly and cumulative $N_2O$ and $CO_2$ fluxes

Weekly $N_2O$ emissions were significantly affected by all possible three-way interactions between earthworms, plants, SWC and time (Table 1 and Fig 2A). $N_2O$ emissions were higher after each watering (with the measurements always being performed 24 h after watering) and were the highest in the second week of the experiment (0.10 g N–$N_2O$ m$^{-2}$ day$^{-1}$, Fig 2A). The intensity and duration of these emission peaks depended on the earthworm and plant treatments but varied with time, as indicated by the significant Ew×Plant×Week interaction. For example, $N_2O$ emissions peaked ten days after the first watering in the presence of anecic earthworms compared to the presence of endogeic earthworms and reached a maximum in the absence of plants. Cumulative $N_2O$ emissions over the 12 weeks of the experiment were the highest in the control without earthworms or plants (0.135 g m$^{-2}$, Fig 2B and 2C). Relative to the control, the cumulative $N_2O$ emissions were 17.0%, 34.6% and 44.8% lower in the anecic, both and endogeic treatments, respectively, and 19.8% lower in the presence of plants (Fig 2C). The Ew×SWC interaction indicates that cumulative $N_2O$ emissions increased with average SWC for the control, anecic and both earthworm treatments but decreased with SWC

**Table 1. Time series and cumulative statistic table.**

| Source | SWC | Litter | $N_2O$ | $cN_2O$ | $CO_2$ | $cCO_2$ |
|---|---|---|---|---|---|---|
| Ew | 46.68*** | 182.89*** | 82.97*** | 7.07*** | 2.53[+] | 1.25 |
| Plant | 173.83*** | 2.91[+] | 34.58*** | 3.82[+] | 5.98* | ns |
| Week | 2758.70*** | 68.85*** | 112.9*** | NA | 230.28*** | NA |
| SWC | NA | NA | 0.04 | 6.35* | 128.85*** | 7.54** |
| Ew×Plant | 2.63[+] | 3.65* | 9.52*** | 2.05 | 2.01 | ns |
| Ew×Week | 9.40*** | 198.63*** | 5.61*** | NA | 1.97** | NA |
| Plant×Week | 24.01*** | ns | 9.32*** | NA | 4.93*** | NA |
| Ew×SWC | NA | NA | 14.84*** | 4.45** | 5.16** | 3.16* |
| Plant×SWC | NA | NA | 1.43 | ns | ns | ns |
| Week×SWC | NA | NA | 8.34*** | NA | ns | NA |
| Ew×Plant×Week | ns | ns | 1.57* | NA | ns | NA |
| Ew×Plant×SWC | NA | NA | 9.51*** | ns | ns | ns |
| Ew×Week×SWC | NA | NA | 3.81*** | NA | ns | NA |
| Plant×Week×SWC | NA | NA | 5.52*** | NA | ns | NA |
| Ew×Plant×Week×SWC | NA | NA | ns | NA | ns | NA |
| $_mr^2$ | 0.81 | 0.91 | 0.67 | 0.49 | 0.43 | 0.27 |

Minimal adequate models for weekly time series (SWC, Litter cover, $N_2O$, and $CO_2$) and cumulative emissions ($cN_2O$ and $cCO_2$) as affected by the earthworm (Ew), plant (Plant), sampling week (Week), soil water content (SWC) and their interactions. "NA" stands for non-applicable, "ns" stands for variables that were not significant ($P > 0.1$) and were not retained in the minimal adequate models whereas $_mr^2$ represents the marginal coefficient of determination. Figures are F-values.

***$P < 0.001$

**$P < 0.01$

*$P < 0.05$

[+]$P < 0.1$.

in the presence of endogeic earthworms (Table 1 and Fig 4A). $CO_2$ weekly emissions were significantly affected by the Ew×Week, Plant×Week and Ew×SWC interactions and were higher after rewatering at higher SWC (Table 1 and Fig 3A). The cumulative $CO_2$ emissions showed no significant response to earthworm or plant presence (Fig 3B and 3C). However, a significant interactive effect of earthworms with SWC was found (Table 1 and Fig 4E), indicating that when SWC was relatively high (> 85.9% SWC on average), as in the endogeic and earthworm-control treatments, cumulative $CO_2$ emissions generally decreased with increasing SWC. The opposite was true in the presence of anecic species, while no relationship was found in the presence of both species. Additionally, when the SWC effect was not included as a predictor, the cumulative $CO_2$ emissions relative to the control were 5.9% and 11.4% lower in the endogeic and both earthworm treatments, respectively, as indicated by Tukey's HSD post hoc test (Fig 3B). Conversely, plant presence increased the cumulative $CO_2$ emissions by 6%.

## Plant, litter, microbial activity, nutrients and porosity

Although not significant, the presence of earthworms led to higher aboveground plant biomass compared to the control (+57%, +25% and +41% for anecic, endogeic and both species, respectively, Table 2 and S4A Fig). Litter cover depended on the presence of earthworms and varied with time and plant presence, as indicated by the Ew×Week and Ew×Plant interactions, respectively (Table 1 and Fig 1C). After 12 weeks, anecic earthworms reduced litter cover to 5% and 6.4% in the absence and presence of plants, respectively. Moreover, when both species were present, litter cover was 40% without plants and 15% with plants (S4B Fig). At week 12, SWC was positively correlated with litter cover (r = 0.55, t = 4.86, P < 0.001, n = 56; S3 Fig),

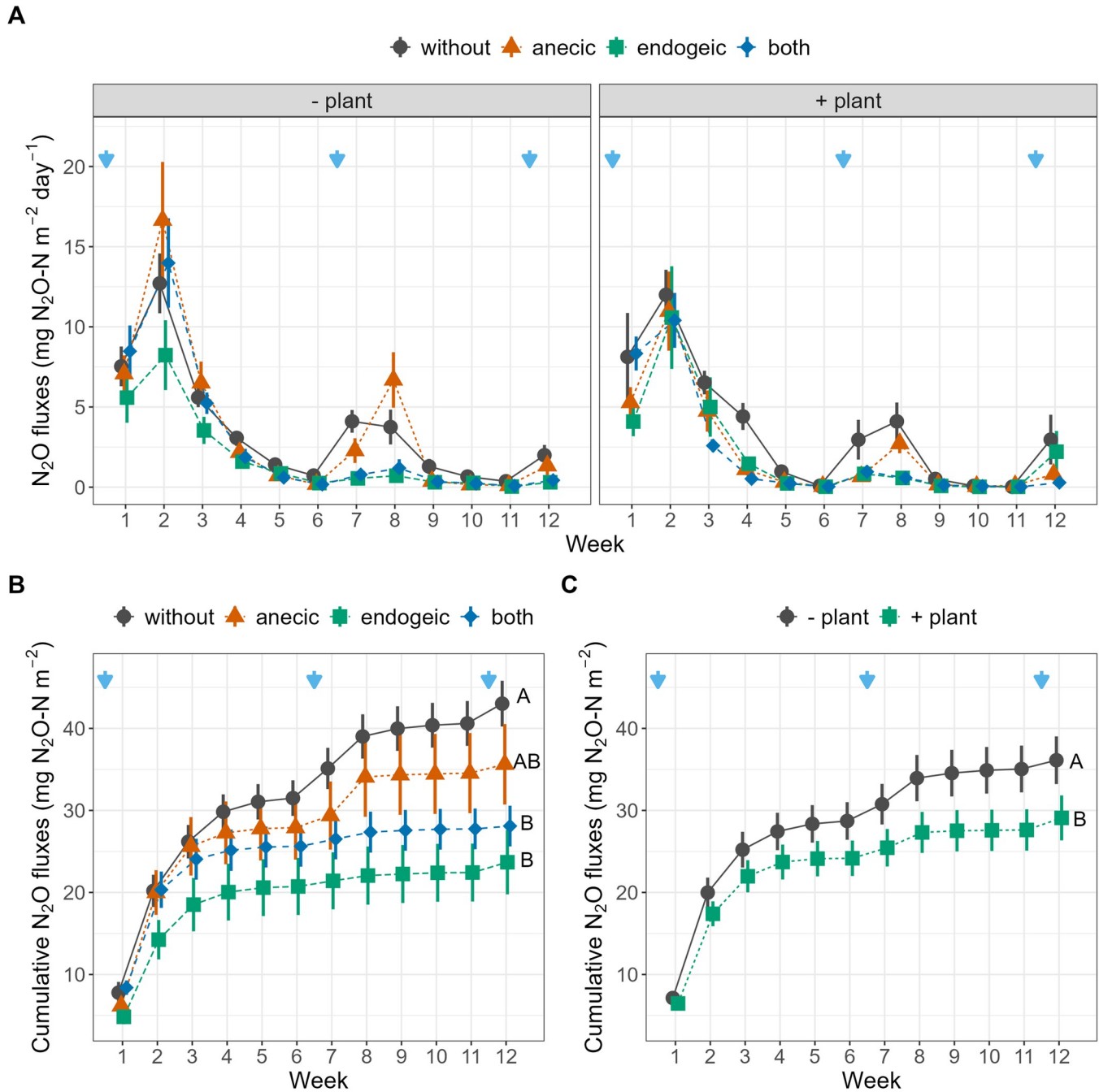

**Fig 2. N$_2$O emissions during the 12-week experiment.** (A) Weekly (B, C) and cumulative N–N$_2$O emissions as affected by the earthworm and plant treatments. Blue arrows represent watering events. Error bars represent ± 1 SEM. Different letters represent significantly different levels as estimated by Tukey's HSD post hoc test. The effects of earthworm treatment independent of plant treatment, and vice-versa, are displayed when the Ew×Plant interaction is not significant (B, C).

suggesting that, at least in part, the presence of earthworms (especially the anecic *L. terrestris*) also affected SWC via higher evaporation from bare soil due to litter burial. We found marginally higher denitrification potential in the presence of anecic earthworms (+22%) relative to the control but no plant effects (Table 2 and S4C Fig). The soil nitrate content (NO$_3^-$) was

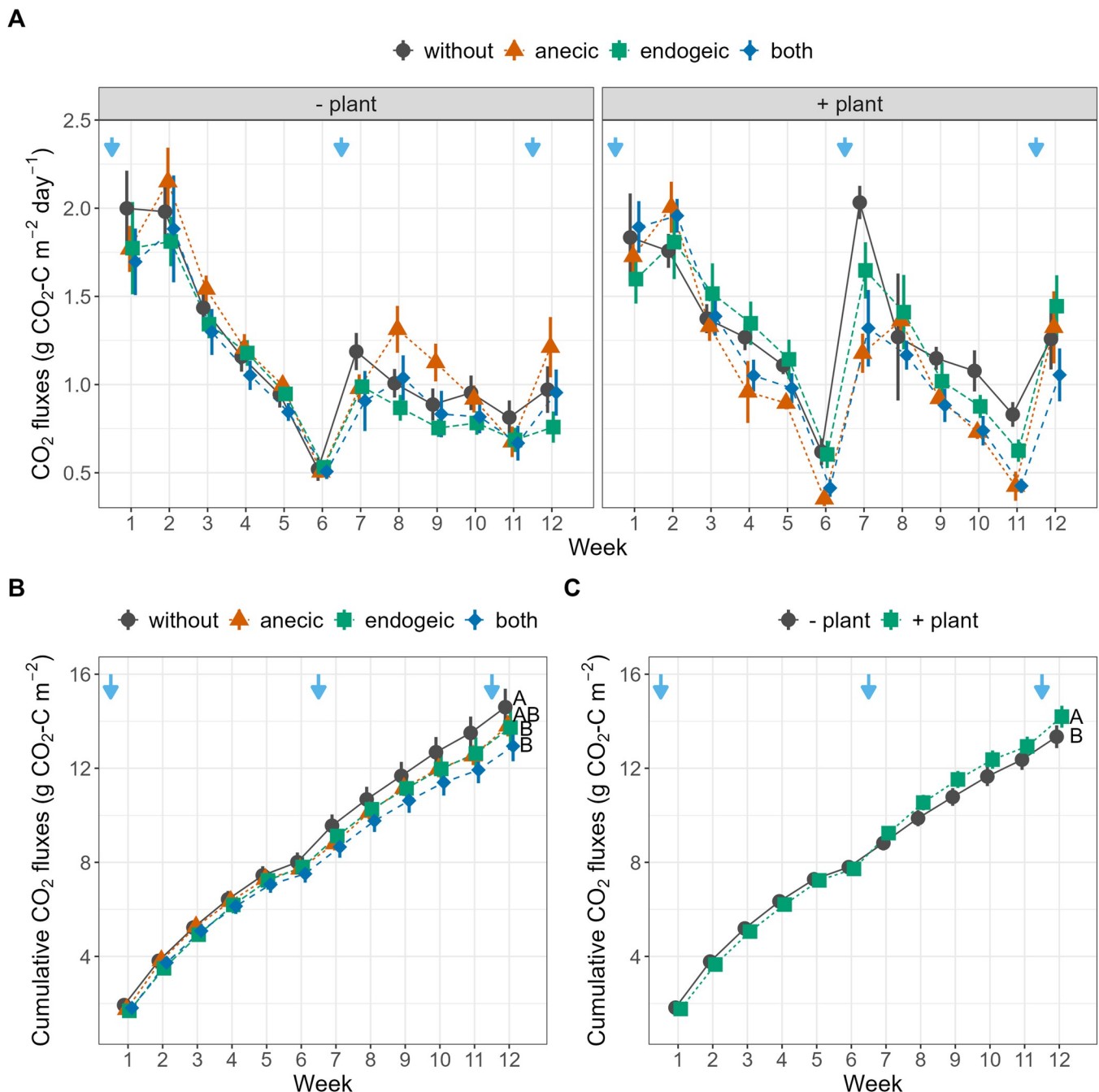

**Fig 3. $CO_2$ emissions during the 12-week experiment.** (A) Weekly (B, C) and cumulative $C-CO_2$ emissions as affected by the earthworm and plant treatments. Blue arrows represent watering events. Error bars represent ± 1 SEM. Different letters represent significantly different levels as estimated by Tukey's HSD post hoc test. The effects of earthworm treatment independent of plant treatment, and vice-versa, are displayed when the Ew×Plant interaction is not significant (B, C).

always lower (-80%) in the presence of plants and increased in the presence of earthworms, with synergistic effects in the presence of both species (Table 2 and S4G Fig). The macropore volume in the topsoil layer (L1) was not affected by any experimental treatment (Table 3 and S2 Fig). We found lower macropore volume in the presence of plants in the other three layers

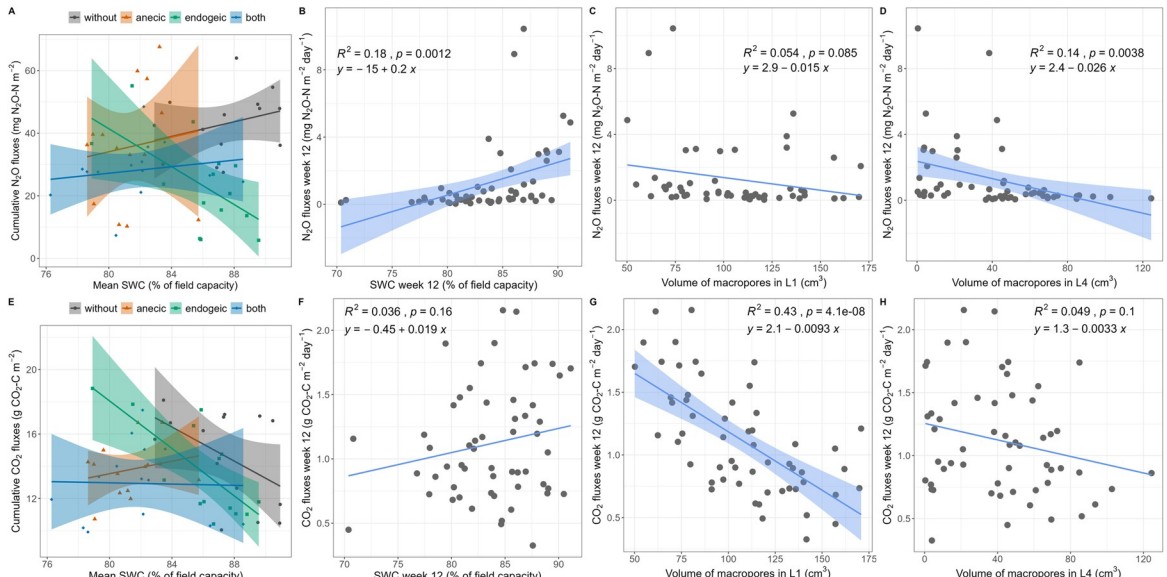

**Fig 4. N₂O and CO₂ emissions as affected by soil water content and macroporosity.** (A, E) Total cumulative emissions as affected by the Ew×SWC interaction, where SWC represents the 3-month average SWC. (B, C, D, F, G, H) Relationships between $N_2O$ and $CO_2$ emissions measured in the last experimental week (week 12), (C, F) soil water contents and (D,G) X-ray tomography estimated volumes of macropores in the topsoil layer (L1) and (D, H) in the bottom layer for $N_2O$ (L4). When significant, the linear regression line and 95% confidence intervals are displayed along with the regression equation, coefficient of determination ($R^2$) and p value. Note that this relationship may differ from the mixed-effect model results (Tables 1 and 4).

(-26.0% in L2, -23.8% in L3, -18.1% in L4). Macropore volumes in L2–L4 were affected by the earthworm treatment, with the highest volume in the mesocosms with endogeic earthworms, followed by both the anecic and control treatments (S2 Fig). Similar trends were observed when macropores were differentiated into burrows and cracks (S1 Table and S2 Fig). Burrow volume was largely driven by the earthworm treatment, with high coefficients of determination in L2, L3, and L4 and in total (i.e., the sum of L1 to L4). Plant presence only affected the total burrow volume and that in L2 (S1 Table).

## Exploration of multiple predictors for final N₂O and CO₂ fluxes

Out of the 16 tested potential predictors (S3 Fig), the MCP-penalized multiple regression for $N_2O$ emissions indicates that, in addition to a retained positive coefficient for SWC ($4.663e^{-04}$), the macropore volumes in the first and fourth soil layers (Vpores_L1 and Vpores_L4) were also retained with negative coefficients ($-1.01e^{-05}$ and $-4.18e^{-05}$, respectively) at minimum cross-validation error with lambda = 0.001 (Fig 4A–4D). $CO_2$ emissions were influenced by macropore volume in the topsoil (Vpores_L1), with a negative coefficient (- 0.034) at minimum cross-validation error with lambda = 0.321 (Fig 4E–4H). Notably, these selected predictors also ranked among those with the highest correlation coefficients with $CO_2$ and $N_2O$ fluxes, as shown by the univariate correlations (S3 Fig). In the final step, we examined how the inclusion of porosity metrics influenced the model performance in explaining the emissions from the last experimental week (Table 3). For $N_2O$ emissions, Vpores_L1 and Vpores_L4 could not be incorporated together due to overfitting and lack of convergence, prompting us to run separate models for each porosity variable.

Minimal adequate models for $N_2O$ emissions that included Vpores_L1 explained more variation than without the porosity metric ($r^2m = 0.86$, vs. $r^2m = 0.69$, respectively) and retained two additional interactions, notably the Ew×SWC×Vpores_L1 three-way interaction with a

**Table 2. Covariables summary statistics table.**

| Plant | Earthworm | Total shoot biomass | Litter cover | DEA | Cmic | BR | Met_Q | NO₃⁻ | NH₄⁺ |
|---|---|---|---|---|---|---|---|---|---|
| | | dry g | % | g N g soil$^{-1}$ h$^{-1}$ | g C-CO$_2$ g soil$^{-1}$ h$^{-1}$ | g C-CO$_2$ g soil$^{-1}$ h$^{-1}$ | Ratio | mg kg$^{-1}$ | mg kg$^{-1}$ |
| - plant | without | | 100 ± 5.6 c | 0.04 ± 0.01 ab | 56.9 ± 2.5 a | 0.8 ± 0 a | 0.5 ± 0 a | 31.7 ± 5.9 abc | 0.4 ± 0.1 a |
| + plant | without | 3.6 ± 0.8 a | 100 ± 6.5 c | 0.05 ± 0.01 ab | 62.3 ± 2.9 a | 0.8 ± 0.1 a | 0.5 ± 0 a | 12.9 ± 6.8 a | 0.3 ± 0.1 a |
| - plant | anecic | | 5 ± 6 a | 0.07 ± 0.01 ab | 63 ± 2.6 a | 0.8 ± 0.1 a | 0.5 ± 0 a | 54.6 ± 6.3 cd | 0.4 ± 0.1 a |
| + plant | anecic | 5.6 ± 0.8 a | 6.4 ± 6 a | 0.05 ± 0.01 ab | 58.9 ± 2.6 a | 0.9 ± 0.1 a | 0.5 ± 0 a | 32.4 ± 6.3 abc | 0.2 ± 0.1 a |
| - plant | endogeic | | 100 ± 6 c | 0.04 ± 0.01 a | 54.4 ± 2.6 a | 0.8 ± 0.1 a | 0.6 ± 0 a | 49.1 ± 6.3 bcd | 0.2 ± 0.1 a |
| + plant | endogeic | 4.5 ± 0.8 a | 100 ± 6 c | 0.05 ± 0.01 ab | 59 ± 2.6 a | 0.8 ± 0.1 a | 0.5 ± 0 a | 23.1 ± 6.3 ab | 0.2 ± 0.1 a |
| - plant | both | | 40 ± 6.5 b | 0.06 ± 0.01 ab | 59.7 ± 2.9 a | 0.8 ± 0.1 a | 0.5 ± 0 a | 65.9 ± 6.8 d | 0.2 ± 0.1 a |
| + plant | both | 5 ± 0.7 a | 15 ± 5.6 ab | 0.07 ± 0.01 b | 61.3 ± 2.5 a | 0.9 ± 0 a | 0.5 ± 0 a | 39.2 ± 5.9 abcd | 0.4 ± 0.1 a |
| | Source | Statistical results (minimal adequate models) | | | | | | | |
| | Ew | ns | 74.39*** | 2.75⁺ | ns | ns | ns | 15.95*** | 2.22⁺ |
| | Plant | NA | 0.93 | ns | ns | ns | ns | 46.41*** | ns |
| | Ew×Plant | NA | 3.08* | ns | ns | ns | ns | ns | ns |
| | $_m$r$^2$ | 0.0 | 0.91 | 0.14 | 0.00 | 0.00 | 0.00 | 0.73 | 0.11 |

Summary statistics (mean ± standard error) and minimal adequate models for the covariables considered as predictors for the $N_2O$ and $CO_2$ emissions from the last week of the experiment in addition to the experimental treatments. Different letters represent significantly different treatments according to Tukey's HSD post hoc test. F-values are shown with significance levels:

***$P < 0.001$

**$P < 0.01$

*$P < 0.05$

⁺$P < 0.1$. $_m$r$^2$ = marginal coefficient of determination, Ew = the earthworm treatment, DEA = denitrifying enzyme activity, BR = basal respiration, Met_Q = metabolic quotient.

positive fitted coefficient (Table 4). The second model that included Vpores_L4 had an intermediate amount of explained variation ($r^2m = 0.72$) and only detected one additional marginally significant SWC×Vpores_L4 interaction (p value = 0.054, Table 4). Regarding $CO_2$ emissions, the inclusion of Vpores_L1 largely increased the amount of variance explained ($r^2m = 0.83$ vs. $r^2m = 0.18$, Table 4). The four-way interaction Ew×Plant×SWC×Vpores_L1 was significant with a positive fitted coefficient, indicating higher $CO_2$ emissions in the presence of earthworms (all treatment combinations containing earthworms) and plants under high SWC levels and high macropore volume in L1.

## Discussion

To further advance our understanding of the effects of earthworms on GHG emissions, our study was designed to simultaneously investigate the effects of earthworms, plants, soil moisture fluctuations, and their interactions, with an experimental setup allowing earthworms and plants to affect soil water status and macroporosity. In line with our first hypothesis, we found not only that the presence of earthworms did not increase the $CO_2$ and $N_2O$ cumulative emissions over 12 weeks but also that the presence of the endogeic species *A. icterica* (alone or with the anecic *L. terrestris*) and the presence of plants reduced $N_2O$ cumulative emissions. Furthermore, earthworms, plants, and their interaction modulated SWC fluctuations and jointly affected weekly $N_2O$ and $CO_2$ emissions. Moreover, we found that GHG emissions were partly explained by increased macropore volume in the first soil layer, resulting from earthworm burrowing activity.

By imposing soil moisture fluctuations and allowing earthworms and plants to modulate these fluctuations, our results illustrate how soil water availability controls $N_2O$ and $CO_2$

**Table 3. Covariables summary statistics table.**

| Plant | Earthworm | Vpores_L1 cm$^3$ | Vpores_L2 cm$^3$ | Vpores_L3 cm$^3$ | Vpores_L4 cm$^3$ | Vpores_tot cm$^3$ |
|---|---|---|---|---|---|---|
| - plant | without | 127.1 ± 10.9 a | 79.7 ± 5.9 bc | 39.4 ± 5.7 ab | 9 ± 5.6 a | 255.2 ± 23 abc |
| + plant | without | 98.8 ± 12.6 a | 57.4 ± 6.8 ab | 20.3 ± 6.6 a | 2.1 ± 6.5 a | 178.6 ± 26.6 a |
| - plant | anecic | 96.9 ± 11.7 a | 60.3 ± 6.3 ab | 57.6 ± 6.1 bc | 39.3 ± 6 bc | 254.1 ± 24.6 abc |
| + plant | anecic | 99.3 ± 11.7 a | 45.5 ± 6.3 a | 37.7 ± 6.1 ab | 23.5 ± 6 ab | 206 ± 24.6 ab |
| - plant | endogeic | 118.4 ± 11.7 a | 96.6 ± 6.3 c | 92 ± 6.1 d | 77.3 ± 6 d | 384.2 ± 24.6 d |
| + plant | endogeic | 92.6 ± 11.7 a | 71.8 ± 6.3 abc | 72.2 ± 6.1 cd | 63.5 ± 6 cd | 300 ± 24.6 bcd |
| - plant | both | 105.6 ± 12.6 a | 77.3 ± 6.8 bc | 78.8 ± 6.6 cd | 65.7 ± 6.5 cd | 327.4 ± 26.6 cd |
| + plant | both | 112.5 ± 10.9 a | 55.1 ± 5.9 ab | 63.2 ± 5.7 bc | 53.6 ± 5.6 cd | 284.4 ± 23 abcd |
| | Source | Statistical results (minimal adequate models) | | | | |
| | Ew | ns | 9.04*** | 32.04*** | 191.71*** | 11.91*** |
| | Plant | ns | 26.63*** | 21.36*** | 13.83*** | 15.78*** |
| | Ew×Plant | ns | ns | ns | ns | ns |
| | $_mr^2$ | 0.00 | 0.48 | 0.68 | 0.94 | 0.59 |

Summary statistics (mean ± standard error) and minimal adequate models of the porosity variables considered as predictors for the $N_2O$ and $CO_2$ emissions from the last week of the experiment in addition to the experimental treatments. Different letters represent significantly different treatments according to Tukey's HSD post hoc test. F-values are shown with significance levels:

\*\*\*P < 0.001

\*\*P < 0.01

\*P< 0.05

+P < 0.1. $_mr^2$ = marginal coefficient of determination, Ew = the earthworm treatment, Vpores = total macroporosity in the four different soil layers from L1 (0–8.5 cm) to L4 (25.5–34 cm).

emissions in complex ways [17, 64]. In general, a combination of limited substrate diffusion at very low water content and limited gas diffusion at high water content leads to maximal $N_2O$ emissions (via nitrification and denitrification) and $CO_2$ emissions (via respiration) at intermediate SWC, approximately 75% of the water-filled pore space [22, 23, 65]. In our experiment, SWC varied considerably, with the lowest values in the presence of plants and anecic earthworms, alone or mixed with endogeic earthworms (Fig 1A and 1B). The significant Ew×SWC interaction (Table 1) observed for cumulative and weekly $N_2O$ and $CO_2$ emissions illustrates the SWC optimal value phenomenon, which occurred at approximately 85% of field capacity or 59% of water-filled pore space (Fig 4A and 4B). Indeed, anecic earthworms create large vertical burrows, increasing water infiltration, and bury leaf litter, increasing the proportions of bare soil and water evaporation [40]. This led to a SWC value lower than the optimum SWC value for microbial activity, thus explaining the positive relationship between SWC and respiration for this species. Conversely, the treatment combinations with the endogeic species, similar to the control with no earthworms, maintained higher than optimal SWC values for soil respiration on average, thus explaining the negative slopes of $CO_2$ fluxes with increasing SWC. The presence of both earthworm species led to SWC values that spanned across the optimum, and no clear relationship between SWC and $CO_2$ could be detected. Simultaneously, the presence of *B. distachyon* grass lowered the average SWC compared to the mesocosms without plants (mean ± se = 82.0 ± 0.6% of WHC), and in this soil moisture range, $CO_2$ emissions increased with SWC (S5 Fig). In the absence of *B. distachyon*, the average SWC was higher (86.2 ± 0.6% of WHC), but increasing the SWC lowered $CO_2$ emissions as the range of SWC was beyond the optimum, and presumably, soil respiration was limited by $O_2$ diffusivity under these conditions (S5 Fig). Regarding $N_2O$ emissions, we observed similar patterns in most cases, except for the control group without earthworms, where emissions still exceeded the

**Table 4. Summary statistic table of last week gas fluxes.**

| Source | N$_2$O model | | | CO$_2$ model | |
|---|---|---|---|---|---|
| | without porosity | With pore volume L1 | with pore volume L4 | without pore volume | with pore volume L1 |
| Ew | 15.46*** | 24.51*** | 16.15*** | ns | 2.1 |
| Plant | ns | 1.77 | ns | 6.95* | 15.9*** |
| SWC | 11.93** | 28.25*** | 13.15*** | 4.12* | 14.33** |
| Vpore | NA | 0.63 | ns | NA | 36.98*** |
| Ew×Plant | ns | 1.22 | ns | ns | 1.27 |
| Ew×SWC | 12.19*** | 18.73*** | 12.81*** | ns | 2.86$^+$ |
| Plant×SWC | ns | 2.00 | ns | ns | 0.32 |
| Ew×Vpore | NA | 5.60** | ns | NA | 0.14 |
| Plant×Vpore | NA | 1.04 | ns | NA | 0.15 |
| SWC×Vpore | NA | 0.66 | 3.83$^+$ | NA | 0.49 |
| Ew×Plant×SWC | ns | 1.19 | ns | ns | 1.81 |
| Ew×Plant×Vpore | NA | 2.04 | ns | NA | 1.14 |
| Ew×SWC×Vpore | NA | 11.79*** | ns | NA | 1.39 |
| Plant×SWC× Vpore | NA | 1.95 | ns | NA | 1.94 |
| Ew×Plant×SWC×Vpore | NA | ns | ns | NA | 3.76* |
| $_m$r$^2$ | 0.69 | 0.86 | 0.72 | 0.18 | 0.83 |

Minimal adequate models presenting the results explaining the CO$_2$ and N$_2$O fluxes from the last sampling (week 12) where the soil porosity-related variables were included in the model as potential predictors (compared with the models without the soil-porosity variables). "NA" stands for non-applicable, "ns" stands for variables that were not significant and were not retained in the minimal adequate models and $_m$r$^2$ represents the marginal coefficient of determination. F-values are shown with significance levels:

\*\*\*P < 0.001

\*\*P < 0.01

\*P< 0.05

$^+$P < 0.1.

optimal SWC values observed in the treatments with earthworms. This finding only partially supports the hypothesis of an optimal SWC mechanism and its effect on N$_2$O emissions. These results suggest that while the interactions between earthworms, plants, and SWC strongly influence N2O emissions, other factors likely come into play when earthworms are absent, such as a significantly different soil porosity status.

The inclusion of the soil porosity data revealed that the total volume occupied by soil macropores in the upper soil layer was an important predictor of GHG emissions (with a negative coefficient, Fig 4G). This suggests that increasing porosity/aeration in the topsoil layer (0–8.5 cm) can decrease N$_2$O and CO$_2$ emissions, presumably by reducing the SWC in the upper and most microbially active soil layers. Interestingly, in line with our third hypothesis, porosity in the bottom layer (25.5–34 cm) was a good predictor of N$_2$O emissions (Fig 4D) and was the only variable that was influenced by earthworm species in the same way as cumulative N$_2$O emissions. Indeed, the number of burrows in the deepest layer was higher in the presence of the endogeic *A. icterica* (alone or alongside the anecic species), a species with high affinity for the deepest soil layers [66], and presumably prevented the development of denitrification-stimulating anaerobic sites. The reduction of N$_2$O emissions via increased soil aeration was also previously suggested by several studies [67, 68] but was not explicitly shown to our knowledge. Our results indicate that this effect was more prevalent in the presence of the endogeic species and seemed to be related to the higher number of burrows produced by this species in contrast to the larger but less numerous semipermanent burrows produced by the anecic species (S3 Fig)

[41]. In contrast, plants significantly reduced soil macropore volume, likely due to roots improving soil structure and stability against crack formation through the production of exudates acting as binding agents or by root mechanical engagement with soil aggregates [69, 70].

Our experiment also allows us to discuss the importance of nutrient availability for GHG emissions. The observed 19.8% reduction in $N_2O$ emissions in the presence of plants also occurred, likely in part due to plant N uptake, as we found that the amounts of soil $NO_3^-$ and $NH_4^+$ were 43% and 20% lower, respectively, in the presence of plants (in line with our second hypothesis), independent of earthworm presence. This finding supports our hypothesis that plants can compete with microorganisms for nutrients and therefore limit bulk microbial activity [37], given the importance of nitrogen availability for nitrification and denitrification [71]. The absence of a positive effect of plants on $CO_2$ fluxes (either weekly fluxes or microbial potential activity at final harvest) is surprising, notably because our experimental design only allowed the combined measurements of $CO_2$ originating from heterotrophic and root respiration. This could be explained by either nutrient (nitrate and ammonium) limitations or an overall low plant effect due to the relatively low plant biomass production of *B. distachyon* in our experiment. The soil $NO_3^-$ concentration increased in the presence of anecic and endogeic earthworms and even more so when both ecological categories were present, leading to 2.1- and 3-fold increases relative to the control in mesocosms with and without plants, respectively. These results can be explained by the combined effect of local vertical litter burial by *L. terrestris* and the horizontal redistribution in the extensive burrow system of *A. icterica* as well as the higher nitrogen concentration in earthworm casts compared to bulk soil [26, 72]. The accelerated burial of surface litter by the anecic species therefore likely contributed to the higher $N_2O$ emissions via increased N availability after watering events, but this effect faded with time and soil drying. Simultaneously, this higher nutrient availability likely contributed to higher plant growth in the presence of earthworms [6]. Despite this increase in nutrient availability, the decreased soil moisture due to earthworm burrowing and the formation of cracks in the topsoil still reduced microbial activity and GHG production. Overall, our study supports the idea that investigating the effect of earthworms on GHG emissions requires the use of an experimental setup that includes plants over a sufficiently long period (> 3 months) [7] and allows water availability to fluctuate due to biological activity.

Mesocosm experiments are highly valuable tools for global change research, but care must be taken in interpreting and extrapolating results, and potential caveats of our study must be mentioned [73, 74]. Because soil properties can strongly influence GHG emissions as well as earthworm cast properties [75, 76], we cannot be sure of the transferability of our results to other soil types. Second, as gas diffusion will occur at the soil–air interface, with gases that are more concentrated in the soil moving toward the atmospheric air, where the concentrations are lower, the bottoms of mesocosms should be as airtight as possible while still allowing for drainage. In our case, the holes at the bottom represented 0.4% of the surface area and were obstructed by the table, thus presumably limiting this bias. Future studies could address this issue by placing the mesocosms on a layer of the same experimental soil or using active drainage systems consisting of suction pumps and tubing equipped with valves that can be closed after drainage. We also acknowledge that the size of our mesocosms, although larger than those in many other studies, could have interfered with earthworm burrowing behavior, especially for deep-burrowing anecic earthworms [51]. Furthermore, as only one earthworm species per ecological category was used, it is unknown whether our findings are transferable to other species from the same ecological categories or whether these findings are also valid for epigeic earthworm species that have been reported to also increase $N_2O$ emissions [7]. It must be noted that earthworm ecological categories were not conceptualized to describe a functional role but rather ecological and morphological groups, which can also explain the high

variability of earthworm species effects within the same ecological category [77]. To the best of our knowledge, our study is the first to investigate the link between earthworm-induced macroporosity and greenhouse gas fluxes; however, the size of the mesocosms combined with the imposed drying–rewetting cycles led to the formation of cracks that unfortunately made the analysis more difficult. As cracks are even more unstable than burrows under drying–rewetting cycles [41], they should be avoided if possible or taken into account in analyses in future experiments. Another caveat is that our weekly measurement frequency of $CO_2$ and $N_2O$ fluxes over 12 weeks may have missed daily variations or higher emission peaks following the watering events. However, our measurements still detected peaks after watering (Fig 2), and while stimulation of emissions under the anecic treatment was detectable, for the endogeic species, the $N_2O$ emission rates were consistently lower than those of the control. Finally, our study duration (3 months) was intermediate, as classified by Lubbers et al. 2013, and earthworm densities in two of our three earthworm treatments were higher than natural densities. Future studies should perform longer experiments with natural densities [7], which, as in our case, can prove to be technically challenging.

In conclusion, our study highlights new mechanisms by which earthworms and plants influence soil GHG emissions in an experimental setup integrating earthworm engineering effects on soil water fluxes and soil porosity, two major mechanisms that have been neglected thus far. The presence of earthworms did not increase $CO_2$ and $N_2O$ emissions and revealed that the endogeic earthworm *A. icterica*, a common species present in Europe and North America, even has the potential to reduce $N_2O$ emissions. Our study is an additional step toward a better understanding of the interactions between soil biota and soil physicochemical properties underlying GHG emissions. Future research on these mechanisms would be highly valuable, especially in an agricultural context, as agriculture is the first sector of $N_2O$ emissions [78], and many mitigation practices have been proposed (e.g., reduced tillage or cover crops) [69]. At the same time, these practices also affect earthworm communities, soil porosity, soil compaction, and water infiltration, which interact and affect soil functions [79–82]. Future research should address these points in experimental setups where the earthworm engineering effect on soil water status and aeration is allowed to take place in a realistic manner.

## Supporting information

**S1 Fig. Schematic depicting the mesocosms dimensions and the elements (base collar and static chamber) used for measuring the $N_2O$ and $CO_2$ emissions.** L1 (0–8.5 cm), L2 (8.5–17 cm), L3 (17–25.5 cm) and L4 (25.5–34 cm depth) represent the four different soil layers that were separately analyzed for soil porosity variables.
(TIF)

**S2 Fig. A)** The effects of treatments on macroporosity volume differentiated as earthworm burrows, cracks, and total macroporosity (burrows + cracks). Error bars represent ± 1 SEM (L1 for 0–8.5 cm, L2 for 8.5–17 cm, L3 for 17–25.5 cm and L4 for 25.5–34 cm depth; see Fig 1). **B)** Examples of 3D reconstruction of the soil macroporosity differentiated as burrows, cracks and total (burrows + cracks) for the two earthworm species alone.
(TIF)

**S3 Fig. Correlation matrix of all predictors measured in the last week of the experiment.** Ew_bm = Earthworm biomass (g FW mesocosm-1), Ew_no = Number of added earthworms per mesocosm (number), SWC = Soil water content relative to field capacity (% of field capacity), Plant_bm = Aboveground plant biomass (g DW mesocosm$^{-1}$), Litter = Percentage of soil surface covered by litter (%), EA = Potential denitrification enzymatic activity (µg N g-1 soil

DW h-1), Cmic = Microbial biomass C ($\mu$g Cmic g soil $^{-1}$ DW), BR = Microbial basal respiration ($\mu$g C-CO2 g$^{-1}$soil DW h$^{-1}$), Met_Q = Microbial metabolic quotient ($\mu$g C–CO2 $\mu$g$^{-1}$ h$^{-1}$), $NH_4^+$ = Ammonium content in soil at the end of the experiment (mg kg$^{-1}$), $NO_3^-$ = Nitrate content in soil at the end of the experiment (mg kg$^{-1}$), Vpores_L1-4 & tot = Macropore (burrow + cracks) volume estimated from CT scan in the 0–8.5 cm layer (L1), 8.5–17 cm layer (L2), 17–25.5 cm layer (L3), 25.5–34 cm layer (L4), and in the whole mesocosm (tot) (cm$^3$), Vburrows_ L1-4 & tot = Burrow volume estimated from CT scan in the 0–8.5 cm layer (L1), 8.5–17 cm layer (L2), 17–25.5 cm layer (L3), 25.5–34 cm layer (L4), and in the whole mesocosm (tot) (cm$^3$), Vcracks_L L1-4 & tot = Cracks volume estimated from CT scan in the 0–8.5 cm layer (L1), 8.5–17 cm layer (L2), 17–25.5 cm layer (L3), 25.5–34 cm layer (L4), and in the whole mesocosm (tot) (cm$^3$).
(TIF)

**S4 Fig. Effects of experimental treatments on several predictors of $CO_2$ and $N_2O$ fluxes measured at the end of the experiment.** Green = in presence of plant, dark grey = in absence of plant. Different letters represent significantly different treatments according to Tukey's HSD post hoc test. Error bars represent ± 1 SEM.
(TIF)

**S5 Fig.** Cumulative N2O (A) and CO2 (B) emissions as affected by Plant×SWC interaction, where SWC represents the 3-month average SWC and the emissions the total cumulative gas emissions.
(TIF)

**S1 Table. Effects earthworms (Ew) and plant treatments on the total macroporosity volume (pores) as well as differentiated as burrows and cracks (see Table 2 for detailed variable description).** The "ns" abbreviation stands for variables that were not significant and were not retained in the minimal adequate models whereas $_mr^2$ represents the marginal coefficient of determination. ***P < 0.001; **P < 0.01; *P< 0.05; +P < 0.1.
(DOCX)

## Acknowledgments

This study benefited from the CNRS human and technical resources allocated to the ECOTRONS Research Infrastructure and AnaEE France. We thank Thierry Morvan for his assistance with the organization of the soil excavation at the EFELE experimental site as well as Thierry Mathieu and David Degueldre from the technical platform Terrain d'Expériences du C.E.F.E. for their support with the greenhouse and production of gas sampling collars. Microbial and soil analyses were done at Plateforme d'Analyses Chimiques en Ecologie (PACE) supported by the LabEx CeMEB, an ANR "Investissements d'avenir" programme (ANR-10-LABX-04-01).

## Author Contributions

**Conceptualization:** Johanne Nahmani, Alexandru Milcu.

**Data curation:** Pierre Ganault, Yvan Capowiez, Alexandru Milcu.

**Formal analysis:** Pierre Ganault, Alexandru Milcu.

**Funding acquisition:** Johanne Nahmani, Alexandru Milcu.

**Investigation:** Pierre Ganault.

**Methodology:** Pierre Ganault, Yvan Capowiez, Nathalie Fromin, Ammar Shihan, Isabelle Bertrand, Bruno Buatois, Alexandru Milcu.

**Validation:** Pierre Ganault.

**Visualization:** Pierre Ganault, Alexandru Milcu.

**Writing – original draft:** Pierre Ganault, Alexandru Milcu.

**Writing – review & editing:** Pierre Ganault, Alexandru Milcu.

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
