## [Decision Letter · Decision Letter 0]

29 Aug 2023

PONE-D-23-23009Earthworms and plants can decrease soil greenhouse gases emissions by modulating soil moisture fluctuations and soil macroporosity in a mesocosm experimentPLOS ONE

Dear Dr. Ganault,

Thank you for submitting your manuscript to PLOS ONE. After careful consideration, we feel that it has merit but does not fully meet PLOS ONE’s publication criteria as it currently stands. Therefore, we invite you to submit a revised version of the manuscript that addresses the points raised during the review process.

We look forward to receiving your revised manuscript.

Kind regards,

Farhan Hafeez, Ph.D.

Academic Editor

PLOS ONE

Journal Requirements:

Additional Editor Comments (if provided):

Dear Author(s),

I write to you regarding the manuscript PONE-D-23-23009 entitled "Earthworms and plants can decrease soil greenhouse gases emissions by modulating soil moisture fluctuations and soil macroporosity in a mesocosm experiment" which you have submitted to PLOS One.

The reviewers have now commented on the manuscript. You will see that they are advising substantial revision of the manuscript. In addition to the reviewers’ comments, the overall write-up and the figures’ quality needs considerable improvement. When revising your work, please submit a list of changes or a rebuttal against each point being raised through track changes mode or by using bold or colored text.

Please revise the manuscript strictly according to the appended comments from the reviewers. While revising, need is to double check that all the references cited within the text have corresponding references.

Sincerely,

Reviewers' comments:

Reviewer's Responses to Questions

**Comments to the Author**

1. Is the manuscript technically sound, and do the data support the conclusions?

Reviewer #1: Yes

Reviewer #2: Partly

2. Has the statistical analysis been performed appropriately and rigorously? 

Reviewer #1: Yes

Reviewer #2: I Don't Know

3. Have the authors made all data underlying the findings in their manuscript fully available?

Reviewer #1: Yes

Reviewer #2: Yes

4. Is the manuscript presented in an intelligible fashion and written in standard English?

Reviewer #1: Yes

Reviewer #2: Yes

5. Review Comments to the Author

Reviewer #1: Ganault et al. present data from a mesocosm experiment in which they investigated the combined effects of earthworms and plants on soil moisture, porosity, and CO2 and N2O emissions. The authors conclude that earthworms reduced greenhouse gas emissions via effects on soil porosity and thus moisture. The novelty of the study is the use of mesocosms in which both plants were present and earthworms could move more freely than in comparable laboratory experiments. The study is timely and written well. I provide some comments below on points that were unclear to me or could be improved.

Introduction

General: The intro is very much focused on N2O. I would like to see more information on how fluctuations in soil moisture, root exudates by plants, and earthworms can influence C stability and CO2 emissions from soils and on the mechanisms behind.

L53/54 That earthworms increase CO2 and N2O emissions is far from being an established fact. I thus suggest using subjunctive here and/or citing other studies that found no changes in or reduced greenhouse-gas emissions in the presence of earthworms.

L56 ff Stabilization of C and/or N in soil is another important process affecting GHG emissions, which could/should be added here with appropriate references.

L59 delete "of"

L63 Reference 13 is a case study and likely not sufficiently representative to support the statement made here. I suggest adding references.

L97 ff In terms of the earthworm effects on water flows, it might be worth noting that anecics create mainly vertical burrows, while those of endogeics are mostly horizontal.

L124 Hypothesis 3) is not entirely clear. How do interactions between the ecological categories (if this is referred to here) affect GHG emissions?

Methods

L162 Were these inputs comparable to those the soils receive under natural conditions?

L163 This appears to be a wild mixture of litter inputs. Wouldn't it be more realistic to use litter from the species that was grown in the mesocosms? Why were these litter species chosen?

L170 Why was the earthworm biomass 2- to 3-fold higher in the mesocosms than under natural conditions?

L176 How many replicates were there in total?

L223 Not clear what the authors refer to here with "the two block".

L264 I wonder how this visual estimation looked like, how precise it can be, and what 5% intervals are. Maybe the authors could provide more info on this.

Results

L393 ff Again, how exactly was litter cover assessed? Was there a standard approach for this? I cannot imagine that simply looking at the litter can yield sound results.

L423 Better "model performance"

Discussion

L448 Better "cumulative N2O emissions"

L486 "burrows"; in general, the authors should recheck their manuscript for typos and grammar.

L495 "root"; it is unclear why aggregate stability should be related to a reduced macropore volume; moreover, there is nothing about aggregates or pore volume in the cited reference.

L517 ff But the current study is short-term as well.

L521 Do the authors have references for this "common practice"?

L522 ff I appreciate the fact that the authors included a limitations paragraph in their manuscript. Points that could be added are that their experiment was relatively short-term and that earthworm densities strongly exceeded those commonly found in the field.

L549/550 In the text above, the authors state that measurement of emissions after watering events leads to bias. The statements here and above are somewhat contradictory.

Fig. 4 All panels. I suggest not plotting trend lines for non-significant regressions. It is then easier for the reader to assess significant and non-significant results.

Fig. S5. Not clear whether relations are significant or not.

Reviewer #2: Th manuscript “Earthworms and plants can decrease soil greenhouse gases emissions by modulating soil moisture fluctuations and soil macroporosity in a mesocosm experiment” reports the results of greenhouse mesocosm experiment. The authors have shown that presence of earthworms in soil greatly reduced the emissions of greenhouse gases.

The quality of the figures is not good and needs to be improved. Legends cannot be read since in most figures they are blurred. The figures must be revised with better quality. Moreover, figure number and their caption are missing which makes it difficult to follow results and their interpretation.

English of the manuscript needs to be improved. It is suggested to authors to get proofread by native English speaker to get rid of small mistakes. For instance, in methodology (Line 133-136) and at many other places authors switch between past and present tense. It is advisable to present this section in past tense.

It is difficult to follow the results of tomographic analyses. It is strongly suggested to clearly cite the figures as well as add figure captions.

6. PLOS authors have the option to publish the peer review history of their article (what does this mean?). If published, this will include your full peer review and any attached files.

Reviewer #1: No

Reviewer #2: **Yes: **Sohaib Aslam

---

## [Author Response · Author response to Decision Letter 0]

10 Oct 2023

We sincerely thank the reviewers for the constructive comments on our manuscript and we thank the editor for the opportunity to provide a revised version. We made a point by point response to the reviewers’ comments and indicated the subsequent changes in the manuscript. We were able to include most of the reviewers suggestions, especially the requirement of more details in the introduction by reviewer 1, while avoiding to make the manuscript too much longer. We carefully checked the writing to avoid typos and made sure the figures are available in high resolution with a detailed caption. 

We are truly grateful for the reviewers’ great work and numerous relevant comments that helped improve the manuscript.

---

## [Decision Letter · Decision Letter 1]

7 Nov 2023

PONE-D-23-23009R1Earthworms and plants can decrease soil greenhouse gases emissions by modulating soil moisture fluctuations and soil macroporosity in a mesocosm experimentPLOS ONE

Dear Dr. Ganault,

Thank you for submitting your manuscript to PLOS ONE. After careful consideration, we feel that it has merit but does not fully meet PLOS ONE’s publication criteria as it currently stands. Therefore, we invite you to submit a revised version of the manuscript that addresses the points raised during the review process.

We look forward to receiving your revised manuscript.

Kind regards,

Farhan Hafeez, Ph.D.

Academic Editor

PLOS ONE

Journal Requirements:

Additional Editor Comments:

Dear Author(s),

I write to you regarding the manuscript PONE-D-23-23009R1 entitled "Earthworms and plants can decrease soil greenhouse gases emissions by modulating soil moisture fluctuations and soil macroporosity in a mesocosm experiment" which you have submitted to PLOS One.

The reviewers have now commented on the manuscript. You will see that they are advising minor revision of the manuscript. When revising your work, please submit a list of changes or a rebuttal against each point being raised through track changes mode or by using bold or colored text.

Please revise the manuscript strictly according to the appended comments from the reviewer(s). While revising, need is to double check that all the references cited within the text have corresponding references.

Sincerely,

Reviewers' comments:

Reviewer's Responses to Questions

**Comments to the Author**

1. If the authors have adequately addressed your comments raised in a previous round of review and you feel that this manuscript is now acceptable for publication, you may indicate that here to bypass the “Comments to the Author” section, enter your conflict of interest statement in the “Confidential to Editor” section, and submit your "Accept" recommendation.

Reviewer #1: All comments have been addressed

Reviewer #2: All comments have been addressed

2. Is the manuscript technically sound, and do the data support the conclusions?

Reviewer #1: Yes

Reviewer #2: (No Response)

3. Has the statistical analysis been performed appropriately and rigorously? 

Reviewer #1: Yes

Reviewer #2: Yes

4. Have the authors made all data underlying the findings in their manuscript fully available?

Reviewer #1: Yes

Reviewer #2: Yes

5. Is the manuscript presented in an intelligible fashion and written in standard English?

Reviewer #1: Yes

Reviewer #2: (No Response)

6. Review Comments to the Author

Reviewer #1: The authors mostly well addressed my comments. However, I have two further remarks:

1. I suggest using more suitable references at the end of the sentence in L55-58, as neither Cotrufo et al. nor Schmidt et al. specifically refer to how earthworms influence C stability or CO2 emissions. There are newer and more appropriate references for this statement, and I believe the authors are capable of finding and citing these references here. They may also amend this statement by referring to how earthworms influence aggregation, and thus stabilization of SOM and C, or other formation processes of stabilized C, which would better address my original concern.

2. Regarding the litter cover assessment: I suggest providing more information or a justification in the manuscript of why the authors think the method was sound

Reviewer #2: Authors tried to address the comments, however, I still feel there is problem with English language.

7. PLOS authors have the option to publish the peer review history of their article (what does this mean?). If published, this will include your full peer review and any attached files.

Reviewer #1: No

Reviewer #2: No

---

## [Author Response · Author response to Decision Letter 1]

20 Nov 2023

We sincerely thank the reviewers for the additional comments on our revised manuscript and we thank the editor for the opportunity to provide a second revised version. We made a point by point response to the reviewers’ comments and indicated the subsequent changes in the manuscript. We hope that reviewer one will now be satisfied by the justification of the litter visual assessment method and the introduction of earthworm effects on carbon stabilization. We also sent the manuscript to American Journal Expert for language checking.

---

## [Decision Letter · Decision Letter 2]

23 Nov 2023

Earthworms and plants can decrease soil greenhouse gas emissions by modulating soil moisture fluctuations and soil macroporosity in a mesocosm experiment

PONE-D-23-23009R2

Dear Dr. Ganault,

We’re pleased to inform you that your manuscript has been judged scientifically suitable for publication and will be formally accepted for publication once it meets all outstanding technical requirements.

Kind regards,

Farhan Hafeez, Ph.D.

Academic Editor

PLOS ONE

Additional Editor Comments (optional):

Dear Author(s),

I write to you regarding the manuscript PONE-D-23-23009R1 entitled "Earthworms and plants can decrease soil greenhouse gas emissions by modulating soil moisture fluctuations and soil macroporosity in a mesocosm experiment” which you have submitted to PLOS One.

The reviewers have now commented on your revised manuscript. I am pleased to inform you that your manuscript is now sufficiently improved for possible publication in PLOS One. The formal acceptance is subject to fulfillment of all the technical requirements.

On behalf of PLOS One, I appreciate you for your contribution. Please keep us in mind for any future work that you consider to be appropriate for our readers.

Sincerely,

Farhan Hafeez, PhD

Reviewers' comments:

Reviewer's Responses to Questions

**Comments to the Author**

1. If the authors have adequately addressed your comments raised in a previous round of review and you feel that this manuscript is now acceptable for publication, you may indicate that here to bypass the “Comments to the Author” section, enter your conflict of interest statement in the “Confidential to Editor” section, and submit your "Accept" recommendation.

Reviewer #1: All comments have been addressed

2. Is the manuscript technically sound, and do the data support the conclusions?

Reviewer #1: Yes

3. Has the statistical analysis been performed appropriately and rigorously? 

Reviewer #1: Yes

4. Have the authors made all data underlying the findings in their manuscript fully available?

Reviewer #1: Yes

5. Is the manuscript presented in an intelligible fashion and written in standard English?

Reviewer #1: Yes

6. Review Comments to the Author

Reviewer #1: (No Response)

7. PLOS authors have the option to publish the peer review history of their article (what does this mean?). If published, this will include your full peer review and any attached files.

Reviewer #1: No

---

## [Editor Report · Acceptance letter]

13 Dec 2023

PONE-D-23-23009R2 

PLOS ONE

Dear Dr. Ganault, 

I'm pleased to inform you that your manuscript has been deemed suitable for publication in PLOS ONE. Congratulations! Your manuscript is now being handed over to our production team.

Kind regards, 

on behalf of

Dr. Farhan Hafeez 

Academic Editor

PLOS ONE